# Microbially induced calcite precipitation using *Bacillus velezensis* with guar gum

**Rashmi Dikshit[1], Animesh Jain[1], Arjun Dey[2], Aloke Kumar[1]\***

**1** Department of Mechanical Engineering, Indian Institute of Science, Bangalore, India, **2** Thermal Systems Group, U. R. Rao Satellite Centre (formerly ISRO Satellite Centre), Indian Space Research Organisation, Bangalore, India

\* alokekumar@iisc.ac.in

## Abstract

Mineral precipitation via microbial activity is a well-known process with applications in various fields. This relevance of microbially induced calcite precipitation (MICP) has pushed researchers to explore various naturally occurring MICP capable bacterial strains. The present study was performed to explore the efficiency of microbially induced calcite precipitation (MICP) via locally isolated bacterial strains and role of guar gum, which is a naturally occurring polymer, on the MICP process. The strains were isolated from local soil and screened for urease activity Further, the urease positive strain was subjected to urea and calcium chloride based medium to investigate the efficacy of isolated strain for microbial induced precipitation. Among screened isolates, the soil bacterium that showed urease positive behaviour and precipitated calcium carbonate was subjected to 16S rRNA gene sequencing. This strain was identified as *Bacillus velezensis*. Guar gum—a natural polymer, was used as a sole carbon source to enhance the MICP process. It was observed that the isolated strain was able to breakdown the guar gum into simple sugars resulting in two-fold increase in calcium carbonate precipitate. Major bio-chemical activities of isolated strain pertaining to MICP such as ammonium ion concentration, pH profiling, and total reducing sugar with time were explored under four different concentrations of guar gum (0.25%, 0.5%, 0.75% and 1% w/v). Maximum ammonium ion concentration (17.5 µg/ml) and increased pH was observed with 1% guar gum supplementation, which confirms augmented MICP activity of the bacterial strain. Microstructural analysis of microbial precipitation was performed using scanning electron microscopy (SEM) and X-ray diffraction (XRD) techniques, which confirmed the presence of calcium carbonate in different phases. Further, XRD and SEM based studies corroborated that guar gum supplemented media showed significant increase in stable calcite phase as compared to media without guar gum supplementation. Significant diverse group of nitrogenous compounds were observed in guar gum supplemented medium when subjected to Gas Chromatography–Mass spectrometry (GC-MS) profiling.

**Data Availability Statement:** All relevant data are within the paper and its Supporting Information files.

**Funding:** RD received a grant (BT/PR31844/BIC/ 101/1206/2019) from the Department of

Biotechnology, Ministry of Science and Technology, Govt. of India http://dbtindia.gov.in/ & Indian Space Research Organization (ISRO). The funders had no role in study design, data collection and analysis, decision to publish, or preparation of the manuscript.

**Competing interests:** The authors have declared that no competing interests exist.

## Introduction

Mineral precipitation mediated via microbial metabolic activity known as bio-mineralization [1] is a pervasive phenomenon on our planet. Reactions such as sulfate reduction, methane oxidation [2], photosynthesis [3], and urea hydrolysis [4] help in either increasing the environmental pH or metabolization of dissolved inorganic carbon [5]. Microbial induced calcium carbonate precipitation (MICP), one of the mechanisms of bio-mineralization, is a process which creates a favourable microenvironment for calcium carbonate precipitation by microbes [6]. MICP has potential applications such as prevention of soil erosion [7, 8], enhancing the durability of concrete [9], restorations of cultural and historical assets [10–12] and production of bio-cement [13–15].

Urea hydrolysis is an important pathway for MICP due to its exclusively controlled mechanism to get higher concentration of precipitated carbonates [16]. Diverse bacterial strains such as *Sporosarcina pasteurii* (formerly *Bacillus pasteurii* [17, 18], *Bacillus megaterium* [14], *Helicobacter pylori* [17], *Pseudomonas aeruginosa* [17], etc are known to hydrolyse urea by producing the urease enzyme (urea amidohydrolase; EC 3.5.1.5). *Sporosarcina pasteurii*, an alkaliphilic, non-pathogenic and endospore producing bacteria has been extensively explored for MICP [19–22]. A mutant strain of *S. pasteurii* was also developed to enhance urease activity which showed significant increase in urease activity and $CaCO_3$ precipitation as compared to the wild strain of *Sporosarcina pasteurii MTCC 1761* [20]. Improvement in concrete strength and durability of construction material was reported via ureolysis mediated calcium carbonate precipitation with incubation of *Bacillus megaterium* [23]. Of late, Lambert and Randall [24] has shown that the green brick can be manufactured using human urine. Urine is a rich source of urea and contains nutrients like nitrogen, potassium and phosphorus too [25]. Therefore, recyclable urine can be a potential alternative source of urea for MICP mediated applications. Strains belonging to *Bacillus* genus are suitable for MICP applications as they can tolerate adverse environmental stresses such as alkaline pH [26], higher mechanical forces and dehydrating conditions by forming endospores [5, 27]. Given the relevance of MICP to various applications, discovery of naturally occurring MICP-capable novel strains is likely to be beneficial in near future. In this study, an unexplored soil isolated *Bacillus* strain namely *Bacillus velezensis* was explored for MICP.

In nature, biominerals are mainly embedded in organic matrices made up of macromolecules. These macromolecules can be either proteins or polysaccharides which are known to act as templates for growth and precipitation of bio-minerals [28, 29]. Organic macromolecules along with inorganic minerals have proven to be nucleating sites for mineral precipitation [30–32]. In this study, Guar gum (*Cyamopsis tetragonolobus*, family Leguminosae) which is a green, non-toxic, easily available, cost-effective and biodegradable polysaccharide-based natural polymer [33] was explored for its role on bacterial growth and MICP process. India tops the cultivation of guar producing more than 8,50,000 tons and amounting to 80% of the total guar produced all over the world [34]

The present study was performed to isolate and characterize indigenous bacterial strains for calcium carbonate precipitation with and without guar gum supplementation. This was accomplished by screening of soil isolated bacterial strains for urease activity and precipitation of $CaCO_3$ via ureolytic pathway. Molecular identification of urease positive bacterial strain was performed via 16S rRNA gene sequencing. Microbial physiology with different concentrations of guar gum supplementation was explored to determine the role of guar gum in the process of MICP. Further, GCMS profiling was performed to characterise the metabolites formed with and without guar gum supplementation in the medium.

## Materials and methods

### Bacterial strains and its characterization

Bacterial strains used for present study were isolated from soil and maintained on glycerol stock at -80˚C. Strains were sub cultured on nutrient agar media at 30˚C for 24 hours. Seed culture was prepared by inoculating one bacterial colony with 50 ml nutrient broth media in shaking incubator (BioBee, India) at 120 rpm and 30˚C to get optical density of 0.5 at 600 nm. The strains were named as SI1 (Soil isolate 1), SI2 (Soil isolate 2), SI3 (Soil isolate 3) and SI4 (Soil isolate 4). Biochemical characterization of isolated strains was done by performing biochemical assay such as oxidation-fermentation test, catalase test, Voges-Proskauer test, sugar fermentation, etc with the help of *Bacillus* identification kit (Hi-media, India).

### Screening for urea hydrolysis

Isolated strains were screened for urea hydrolysis by streaking a single bacterial colony on urea-agar column and kept at 30˚C for 24 hours. To observe change in medium pH, phenol red was added as an indicator along with media. Urea-agar plate was prepared by 0.1 g glucose, 0.1 g peptone, 0.5 g NaCl, 0.2 g mono-potassium phosphate, 2 g urea, 0.0012 g phenol red dye and 2 g agar to 100 ml distilled water. All chemicals were procured from Hi-Media, India.

Since the medium pH is an important physical parameter for bacterial growth and MICP, urease positive strain was tested for the tolerance over varied range of pH (6–10) in synthetic media-urea (SMU) which was prepared by 0.1 g glucose, 0.1 g peptone, 0.5 g NaCl, 0.2 g mono-potassium phosphate and 2 g urea in 100 ml of distilled water. Before autoclave, the medium pH was adjusted with 1M HCL for acidic and with 1M NaOH for alkaline pH. Medium was inoculated and kept for incubation at 30˚C for 48 hrs. After defined incubation period, the cell density was measured using UV/Vis spectrophotometer.

### Molecular identification of isolate

Molecular identification of urease positive strain was done by 16S rRNA gene sequencing. Colonies from single streak on the agar plate were scraped and suspended in PBS buffer and centrifuged. The pellet obtained was dispersed in 600 μl of cell lysis buffer (Guanidium isothiocyanate, SDS, Tris-EDTA) and mixed by inverting the vial for 5 minutes and incubated for 10 minutes with gentle mixing till the suspension appeared transparent. The solution was layered on top with 600 μl isopropanol. The two layers were mixed gently till white strands of DNA were seen and the solution became homogenous. The spooled DNA was spun to precipitate DNA at 10,000 rpm for 10 minutes. The air-dried pellet was suspended into 50 μl of 1X Tris-EDTA buffer and incubated for 5 min at 55–60˚C to increase the solubility of genomic DNA. 5 μl of freshly extracted DNA along with 3μl of gel loading dye was loaded onto 1% agarose gel and subjected to electrophoresis. Amplification of 16s rRNA gene was performed at Aristogene Bioscience, Bangalore, India using the standard protocol [35] and following primers [36] were used to proceed with the PCR reaction.

Forward primer: 5′ – AGAGTTTGATCCTGGCTCAG—3′
Reverse primer: 5′ – ACGGCTACCTTGTTACGACTT—3′

### In vitro CaCO₃ precipitation

Urease positive strain was evaluated for in-vitro calcium carbonate precipitation with different calcium sources and guar gum supplementation in SMU media. Calcium chloride, calcium lactate, calcium nitrate and calcium acetate were used as calcium sources to explore their influence on microbial induced precipitation. SMU supplemented with 25mM calcium source was

inoculated and kept for incubation at 30 ˚C. Cell density was observed after 48hrs of incubation using UV/Vis spectrophotometer and blank was set with appropriate control (SMU-C without strain). To determine the influence of guar gum in the process of MICP, synthetic media-urea-calcium chloride-guar gum (SMUCG) was prepared by replacing glucose in SMUC medium with 1% (w/v) guar gum (Urban Platter, India). The role of guar gum in this process was validated by comparing the results with precipitate obtained in SMUC media. The quantification of induced precipitates was performed after 7 days of incubation.

The strain was inoculated in all mentioned media and incubated at 30˚C for 7 days under static condition. After the incubation period, samples were centrifuged (Sorvall™ Legend™ X1 Centrifuge, Thermo Fisher Scientific, Germany) at 4˚C and 5000 rpm for 10 minutes. The supernatant was discarded and precipitates were dried at 37˚C for 12 hours in hot air oven (BioBee, India). Precipitates were weighed using analytical weighing balance (ATX224 Unibloc®, Shimadzu, Japan). These precipitates were further observed using SEM imaging and XRD analysis.

## Gas chromatography mass spectrometry (GCMS) profiling

The GCMS profiling was carried out for both the treatments i.e. SMUCG and SMUC. The metabolites were extracted using methanol after 5 days of incubation. The methanol extracts of bacterial metabolites were dried with liquid $N_2$ before processing for GCMS (GC column: Agilent 7890A MS: 5975C MSD Electron Impact Ionization Mass Analyze equipped with Quadrupole Software). Approximate 1 μL of bacterial extract was injected into the GC inlet. N, O-Bis(trimethylsilyl)trifluoroacetamide (BSTFA) was used as a derivatizing agent to derivatize the compound for the analysis. Obtained fractions were identified based on m/z ratio of their mass spectra and matched with NIST2011 mass spectral library.

## Bacterial physiology under guar gum supplementation

Bacterial activity of isolated urease positive strain was investigated in SMUCG medium supplemented with varying concentration of guar gum (0.25%, 0.50%, 0.75% and 1% w/v) and was compared with SMUC medium. The strain was inoculated and incubated at 30˚C. pH (CyberScan pH meter, Eutech Instruments), ammonium concentration and amount of reducing sugar were measured at different time intervals. Nessler's reagent assay was used to quantify ammonium concentration, by 100 μl of Nessler's reagent (Hi-Media, India) in 3 ml of bacterial sample. Absorbance was measured at 425 nm (using UV/Vis spectrophotometer, Shimadzu, Japan) after 10 minutes of incubation [37]. Total reducing sugar was estimated using DNS protocol [38] by measuring optical density at 540 nm.

## Scanning electron microscopy (SEM)

Bacterial precipitates in different media compositions (SMUC, SMUCG) were observed using SEM (Carl Zeiss AG—ULTRA 55, Germany). The dried precipitates were fixed with 2.5% glutaraldehyde in PBS buffer (pH 7.4) for 30 minutes, followed by centrifugation at 5˚C and 5000 rpm for 5 minutes. The supernatant was removed and sample was washed with PBS buffer to remove fixative. The treated samples were dehydrated with serial dilution using 10, 30, 50, 70, 90 and 100% v/v ethanol. The samples were mounted on carbon tape fixed on aluminium stub and kept in desiccator for 24 hours. Before SEM imaging, the dried samples were coated with gold for 120 sec to avoid charging.

## X-Ray diffraction (XRD) analysis

The precipitated samples were collected, air dried and grinded to fine powder. The fine powder was analysed using a commercial X-ray diffractometer (PANalytical Philips diffractometer) using CuKα (λ = 1.54056 Å) X-ray radiation and thoroughly indexed as per Inorganic Crystal Structure Database (ICSD) library using PANalytical X'PertHighScore Plus pattern analysis software.

## Results and discussion

### Screening and identification of strains

Morphological and biochemical characterisation of all soil isolated strains was performed with the help of *Bacillus* kit. All strains were found to be gram-positive, aerobic, spore-forming and catalase-positive (S1 Table in S1 File). Further, qualitative analysis based on pH indicator (phenol red) for urea degradation of all strains was performed on SMU agar column (S1 Fig in S1 File). Colour change was observed only in case of SI1 (S1a Fig in S1 File) due to enhancement in medium pH while no change was observed in SI2 (S1b Fig in S1 File), SI3 (S1c Fig in S1 File) and SI4 (S1d Fig in S1 File) as compared to control (without organism) (S1e Fig in S1 File). SI1 was subjected to 16S rRNA gene sequencing (Fig 1) and based on DNA–DNA relatedness, values have shown approximately 99% similarity with *Bacillus amyloliquefaciens* and was identified as *Bacillus velezensis* (S2 Table in S1 File). *Bacillus velezensis* (Accession number- MCC 4181) was deposited in publicly accessible culture collection—National Centre for Microbial Resource (NCMR, Pune, India) and gene sequences of isolated strain (accession number MN108152) was submitted in National Center for Biotechnology Information (NCBI), India gene bank. Bacterial mobility was examined on agar columns which was stab inoculated with isolated strain after 24 hrs of incubation and represented in Fig 2a. Further, the shape of bacteria was observed by SEM imaging. The bacteria were rod-shaped as seen in Fig 2b. Medium pH is an important parameter for MICP mediated applications. Isolated strain was subjected for varying range of pH (6–10). It was observed that strain was able to tolerate pH from acidic to alkaline range (Fig 2c) and maximum growth was seen at pH 7. To characterize the bacterial growth on varied range of temperature, strain was inoculated in SMU media and kept for incubation at different temperatures (15, 20, 25, 30, 35, 40 and 45 ˚C) and OD was measured after 24 hrs of each incubation. It was observed that *Bacillus velezensis* was able to grow on the experimented temperature range and optimum growth was observed at 30 ˚C. There was no significant difference observed in OD between 30 and 35 ˚C. Minimum growth was observed at 15 ˚C as presented in S2 Fig in S1 File.

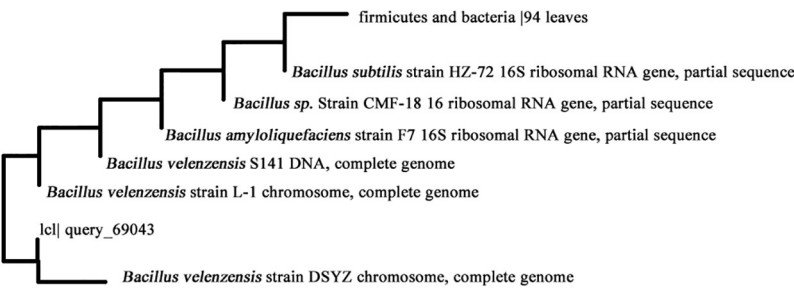

**Fig 1. Phylogenetic tree of 16s rRNA gene sequencing for urease positive strain.**

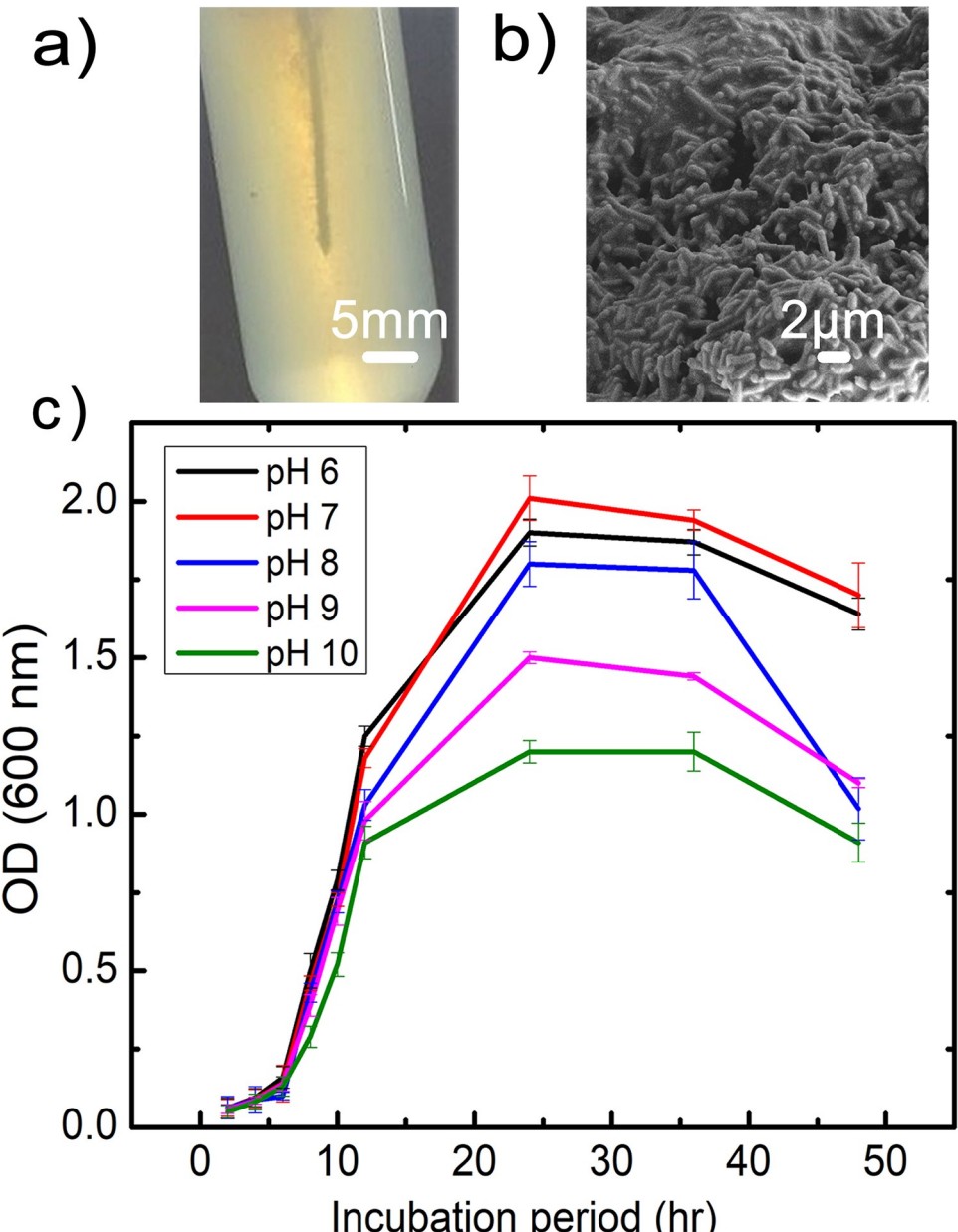

**Fig 2. Characterization of urease positive bacterial strain a) Agar column showing mobility of isolated strain b) SEM image of the bacteria c) temporal evolution of bacterial growth under varied range of pH.**

## Precipitation under flask condition and its microstructural analysis

Utilisation of different calcium source and growth of *Bacillus velezensis* was explored with four different calcium sources, namely, calcium chloride, calcium lactate, calcium nitrate and calcium acetate. The optical density and amount of obtained bacterial induced precipitation is plotted for various calcium sources in S3 Fig in S1 File. Maximum precipitation was observed with calcium lactate and calcium chloride followed by calcium nitrate and calcium acetate (S3 Fig in S1 File). Similar trend was observed with optical density where maximum growth was

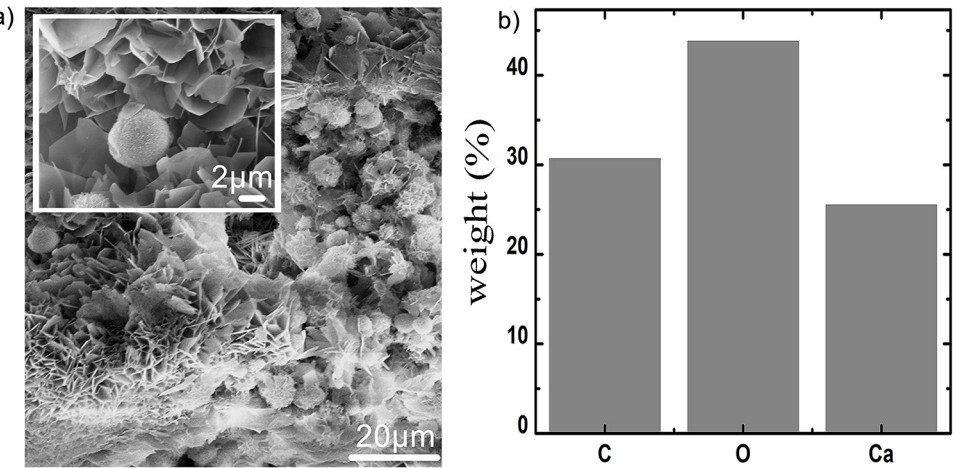

**Fig 3. SEM images of *Bacillus velezensis* mediated precipitated crystals with SMUC medium (a) showing randomly embedded spherical meta-stable calcium carbonate crystals on the bed of flower-petal like nanocrystalline phase (inset: higher magnification image) and (b) elemental distribution of Carbon (C), Oxygen (O) and Calcium (Ca) (Incubation period was 7 days in all cases).**

seen in case of calcium chloride. Subsequently, the experiments were performed with calcium chloride as calcium source.

*Bacillus velezensis* induced precipitates were also investigated with and without guar gum supplementations after 7 days of incubation under flask conditions. Maximum calcite precipitation was observed with guar gum supplemented media (4 g/L) in comparison to SMUC media (1.9 g/L). This indicates that guar gum supports bacterial activity resulting in enhanced amount of precipitation.

Further, microstructural analysis of obtained precipitate was performed using SEM imaging equipped with energy dispersive X-ray spectroscopy (EDS) facility Fig 3a and 3b show the SEM images and EDS data of calcium carbonate precipitates obtained with SMUC treatment while Fig 4a and 4d depict the SEM images with SMUCG treatment. With SMUC, in absence of guar gum supplementation, microbial induced nanocrystalline precipitates were observed (Fig 3a). Several randomly embedded spherical meta-stable vaterite [39, 40] crystals are seen on the bed of nanocrystalline flower-petal like aragonite [40, 41] as shown in Fig 3b. On the other hand, multi-shaped densely populated sharp pointed flaked precipitates were observed in case of SMUCG treatment supplemented with guar gum (Fig 4a). Fig 4b depicts the stacking of stable calcite[39, 42] crystals. The presence of bacteria and its biological activity mediated MICP process was further established in Fig 4c where precipitates sites with numerous holes are found. These observed impressions on the surface of precipitate indicate the presence of bacterially mediated MICP process [6]. The proportion of stable calcite crystals increased after adding guar gum, as seen from the SEM based studies and further confirmed by XRD studies appended later. The precipitates were then subjected to spectral mapping of elements such as carbon, oxygen and calcium to characterize the calcium carbonate precipitation by bacterial strains. The EDS data of SMUC and SMUCG treatments shown in Figs 3b and 4d, respectively indicate the presence of calcium, carbon and oxygen which are the constituents of $CaCO_3$. This was further confirmed by XRD analysis.

Thoroughly indexed XRD patterns of dried and crushed precipitated crystals obtained from both the treatments (SMUC and SMUCG) are shown in Fig 5 (inset)). Detail data of corresponding 2θ, d spacing, hkl, phase and the ICSD reference no. are summarized in S3 Table in S1 File as supplementary material. Fig 5 clearly suggests that different polymorphs of

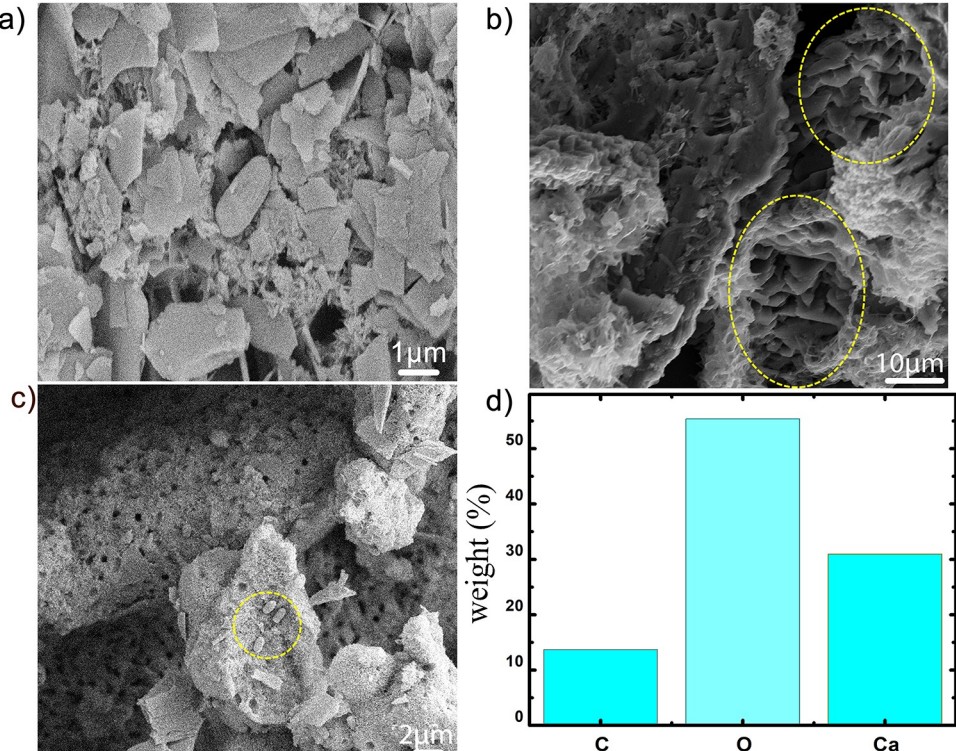

**Fig 4. SEM images of *Bacillus velezensis* mediated precipitated crystals with SMUCG medium (a) showing calcium carbonate precipitates with different shapes, (b) showing stacking of rhombohedral and stable calcite crystals and (c) showing perceptible holes along with bacteria which indicates bacterial induced precipitation and (d) elemental distribution of Carbon (C), Oxygen (O) and Calcium (Ca) (Incubation period was 7 days in all cases).**

calcium carbonate co-exist in both the samples. However, the presence of calcite peaks (Fig 5, inset and S3 Table in S1 File) and their relative intensities (Fig 5) in SMUCG treatment are significantly higher as compared to SMUC treatment. It qualitatively establishes the fact that there is a significant increase in stable calcite phase after guar gum supplementation in the growth medium.

Thus, XRD and SEM analysis clearly reveals that different phases of calcium carbonate were precipitated as a result of difference in bacterial activity under different media supplementation. As significant improvement in precipitation was observed in guar gum as a biopolymer additive medium (SMUCG), its effect was further explored at different guar gum concentrations.

## Bacterial physiology under guar gum supplementation

*Bacillus velezensis* activity was studied with different concentrations of guar gum (0%, 0.25%, 0.50%, 0.75% and 1% w/v) in SMUCG media. Fig 6a shows the plot of ammonium concentration with time. An increase in ammonium concentration was observed until 96 hours in all cases. Further, it decreased till 144 hours and remained approximately constant thereafter. Maximum ammonium concentration was observed with 1% guar gum additive (17.5 µg/ml) followed by 14.9 µg/ml, 13.5 µg/ml, 11.6 µg/ml, 10.3 µg/ml for 0.75%, 0.50%, 0.25% and 0% guar gum additive respectively. It suggests that an increase in guar gum concentration enhances the rate of urea hydrolysis. This is confirmed by the plot of change in pH of growth

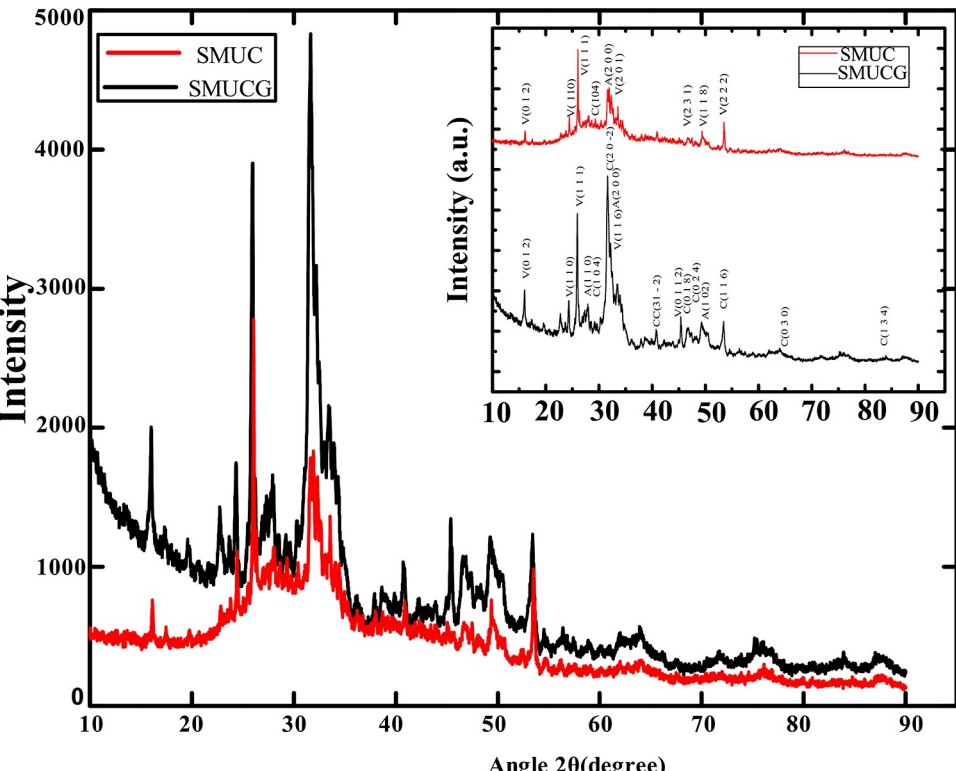

**Fig 5. A comparative view of XRD pattern of *Bacillus velezensis* induced precipitates under guar gum (SMUCG) and without guar gum (SMUC) supplemented media compositions.** Inset figure is showing indexed XRD data with corresponding planes for both the treatments. In the spectrum, V represents vaterite, A for aragonite, C for calcite and CC for calcium carbonate. Incubation period was 7 days in all cases.

medium with time shown in Fig 6a (inset). The experiment was initiated at an acidic pH of 5.5 due to the fact that the media was based on urea and $CaCl_2$ composition and any adjustment in pH towards alkalinity may result in quick abiotic precipitation which is undesirable [6]. An increase in pH is observed until 48 hours in all cases while with higher guar gum

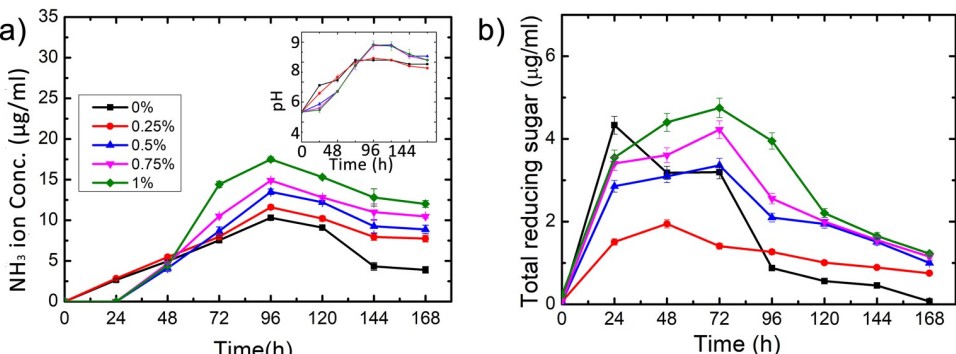

**Fig 6. Exploration of microbial physiology under flask condition with guar gum additive.** (a) temporal evolution of ammonium concentration in SMUCG media and temporal evolution of pH in SMUCG media (inset) (b) change in amount of total reducing sugar in SMUCG media with time compared with SMUC medium (0% curve). Legend shows guar gum percentage (w/v) in SMUCG media (Error bars represents the standard deviation of the data of three independent experiments).

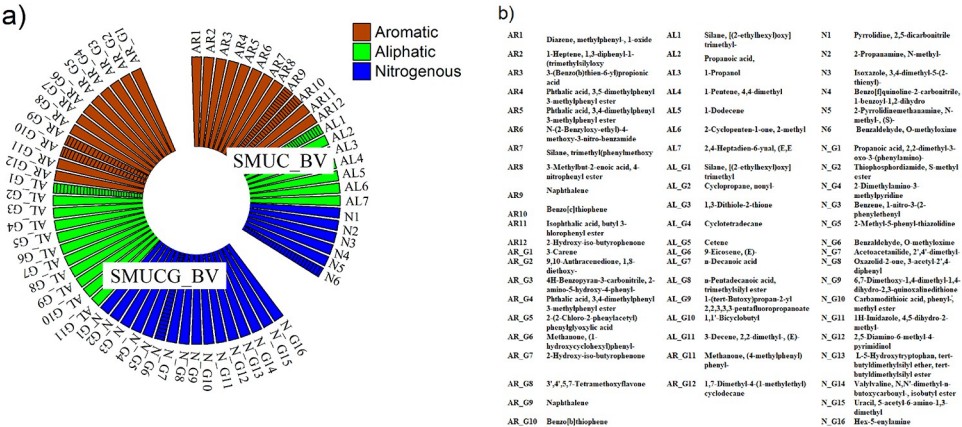

**Fig 7.** (a) GCMS profiling of produced metabolites for guar gum (SMUCG) and without guar gum (SMUC) supplemented media compositions inoculated with *Bacillus velezensis* under flask condition. Incubation period was 7 days in all cases. AR represent aromatic compound, AL for aliphatic compound and N for nitrogenous compound present in medium without guar gum supplementation whereas G represents the above metabolite produced with guar gum supplementation. Hatched column representing common compounds in all treatments. (b) Showing metabolites obtained from GCMS profiling.

concentrations, pH further increased until 96 hours. In the case of 0% and 0.25% guar gum supplemented media, pH reaches to maximum value of 8, whereas, a maximum value of 9 was observed in media supplemented with 0.50%, 0.75% and 1% guar gum.

The effect of guar gum degradation by *Bacillus velezensis* was studied using DNS protocol. In Fig 6b, total reducing sugar is plotted as a function of time. Total reducing sugar content in SMUC media increases till 24 hours followed by a decline. Interestingly, trends for SMUCG media differed from the control sample (SMUC media). For SMUCG media, amount of total reducing sugar in media increases till 48 hours in 0.25% guar gum while it increases till 72 hours in all other guar gum concentrations and decreases thereafter.

In the present study, significant microbial activity was observed in SMUCG treatment compared to SMUC. To substantiate the results established with and without guar gum supplementation and to identify metabolites produced under these conditions, GCMS profiling was carried out. Metabolites thus produced were segregated into nitrogenous, aliphatic and aromatic compounds (Fig 7a and 7b). Distinct metabolites were observed in both the cases. Significant increase in nitrogenous compounds was found in SMUCG (16 compounds) as compared to SMUC (6 compounds). Some common compounds like Naphthalene, Benzo[b] thiophene, Silane, [(2-ethylhexyl)oxy]trimethyl and Benzaldehyde, O-methyloxime were observed in both the treatments. In SMUCG treatment, more fused ring aromatic and nitrogenous compounds such as Methanone, (1-hydroxycyclohexyl)phenyl-, 4H-Benzopyran-3-carbonitrile, 2-amino-5-hydroxy-4-phenyl-, Uracil, 5-acetyl-6-amino-1,3-dimethyl, Hex-5-enylamine, L-5-Hydroxytryptophan, tert-butyldimethylsilyl ether, tert-butyldimethylsilyl ester, Benzene, 1-nitro-3-(2-phenylethenyl were observed as compared to SMUC treatment.

These additional organic macromolecules found in guar gum supplemented medium could be the probable reason for the enhancement of microbial metabolic activity leading to augmentation of the MICP process. However, one nitrogenous compound namely Carbamic acid, N-(2,3-dimethylphenyl)-, oxiranylmethyl ester was observed only in guar gum supplemented medium. Carbamic acid and its derivatives are unstable compounds and carbamic acid is produced as an intermediate compound in urea hydrolysis reaction. Organic macromolecules are known to facilitate the biomineralization phenomenon by regulating its dynamic process

(nucleation site, crystal growth etc) or by providing a structural framework to initiate the process [28–32]. Same observation was made in SMUCG treatment as well.

With increasing research on MICP mediated applications via urease pathway, there is a need to identify more novel species of bacteria with potential for urea hydrolysis. The present study explored the novel urease producing soil bacterium for MICP applications, which was identified as *Bacillus velezensis* through 16S rRNA gene sequencing. Historically, *Bacillus velezensis* was isolated from the river of Velez in Malaga (Southern Spain) and explored by Ruiz-Garcia et al [43]. The important factor for MICP via ureolytic pathway is the selection of strain and its specificity based on the rate of urea hydrolysis. *Bacillus velezensis* has shown potential for urea hydrolysis and degradation of urea was observed after 10 h of inoculation on urea agar plate which indicates its suitability for MICP mediated applications. This novel isolated strain was characterized as a gram-positive bacterium and can tolerate a varied range of pH 5.0–10 and temperature 15 to 45˚C. The precipitation obtained in this study without any additives were identified as polymorphs of calcium carbonate through SEM and XRD results as reported in various studies [6, 17, 44–46]. Major forms of calcium carbonate precipitate are calcite, aragonite, and vaterite of which vaterite and calcite are the most commonly reported polymorphs of bacterial calcium carbonate precipitates [47, 48].

This study lay emphasis on two major outcomes. Firstly, maximum urea hydrolysis was observed with higher concentrations of guar gum as evident from the increase witnessed in ammonium ion concentration and medium pH conducive for MICP. Urease enzyme hydrolyses urea into ammonium ions, increases the pH and thus provides suitable micro-environment to proceed with the reactions [6, 49, 50]. Secondly, the demonstration of capability of *Bacillus velezensis* towards bio-degradation of guar gum into small moiety as evident from the characteristics of total reduced sugar content with time. In case of media with guar gum additive, the bacterial activity was extended up to 72 hours with increased concentration of guar gum. Addition of guar gum also accelerated the rate and enhanced the amount of calcium carbonate precipitation. Moreover, the precipitate consisted of a higher amount of calcite—known to be the most stable form of calcium carbonate [48]. Polymer degradation releases free sugar which could be readily available in the growth medium for bacterial growth and resultant enhancement in the activity. It is reported that the organic matrix covering calcium carbonate precipitation is made up of either proteins or polysaccharides and these macromolecules play an important role in the synthesis of biominerals [51] either by providing a structural framework, or regulating its entire dynamic process such as the nucleation site, growth of crystal and direction of orientation etc. [31, 32, 52, 53]. Positive influence of guar gum on MICP process was also confirmed by the metabolite profiling. Significant diverse group of nitrogenous compounds was produced with the addition of bio-polymer in growth medium, which also justifies that guar gum can be an effective medium supplement to accelerate the MICP process.

## Conclusions

The encouraging results of this study suggest that, this locally available strain could be a model solution for MICP based applications and proposes the use of a biopolymer namely the guar gum as an economical substrate for bacterial cultivation.

## Supporting information

**S1 File.**
(PDF)

## Acknowledgments

Authors would like to acknowledge Dr. Swetha Seshagiri, Associate Professor, Center for Incubation Innovation Research and Consultancy, Jyothy Institute of Technology, Bangalore—560082, Karnataka, India for her kind gift of soil isolates and Ms. Harshita Patangia for her help in performing lab experiments. Authors also acknowledge GCMS Facility at division of Biological Sciences for GCMS analysis and Centre for Nano Science and Engineering (CeNSE) of Indian Institute of Science, Bangalore for SEM and XRD analysis respectively.

## Author Contributions

**Conceptualization:** Rashmi Dikshit, Animesh Jain, Aloke Kumar.

**Data curation:** Aloke Kumar.

**Funding acquisition:** Rashmi Dikshit.

**Investigation:** Animesh Jain.

**Methodology:** Rashmi Dikshit.

**Project administration:** Aloke Kumar.

**Resources:** Rashmi Dikshit.

**Supervision:** Arjun Dey, Aloke Kumar.

**Validation:** Aloke Kumar.

**Writing – original draft:** Rashmi Dikshit, Animesh Jain.

**Writing – review & editing:** Rashmi Dikshit, Animesh Jain, Arjun Dey, Aloke Kumar.

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
