## [Decision Letter · Decision Letter 0]

8 Jun 2020

PONE-D-20-07309

Microbially induced calcite precipitation using Bacillus velezensis with guar gum

PLOS ONE

Dear Dr. Kumar,

Thank you for submitting your manuscript to PLOS ONE. After careful consideration, we feel that it has merit but does not fully meet PLOS ONE’s publication criteria as it currently stands. Therefore, we invite you to submit a revised version of the manuscript that addresses the points raised during the review process.

We look forward to receiving your revised manuscript.

Kind regards,

Arumugam Sundaramanickam, PhD

Academic Editor

PLOS ONE

Additional Editor Comments:

Dear Author,

Reviewers have now commented on your paper. There are a number of queries raised by the reviewers need to be addressed. Please carefully read the reviews comments and modify accordingly. In the revised version you should explain how and where each point of the reviewers' comments has been incorporated.

With regards

A. Sundaramanickam

Academic Editor

2. Please ensure that you refer to Figure 3 in your text as, if accepted, production will need this reference to link the reader to the figure.

Reviewers' comments:

Reviewer's Responses to Questions

**Comments to the Author**

1. Is the manuscript technically sound, and do the data support the conclusions?

Reviewer #1: Yes

Reviewer #2: Yes

Reviewer #3: Yes

2. Has the statistical analysis been performed appropriately and rigorously? 

Reviewer #1: N/A

3. Have the authors made all data underlying the findings in their manuscript fully available?

Reviewer #1: No

4. Is the manuscript presented in an intelligible fashion and written in standard English?

Reviewer #1: Yes

5. Review Comments to the Author

Reviewer #1: In this report, a soil bacterial isolate, Bacillus velezensis, was identified and used to enhance calcite precipitation. In this microbe-induced calcite precipitation, a natural polymer from germinated seeds, guar gum, was supplemented to increase the reaction. The authors showed that 1% guar gum could conduct the best biological activity of B. velezensis, which indicated by ammonium concentration and increased pH value. The microstructure of calcite precipitation was observed by SEM and X-ray diffraction and showed more calcite peaks while added with guar gum in the presence of B. velezensis. In addition, significant diverse groups of nitrogenous compounds were observed in GC-MS profile. Some questions:

1.There are different criteria to evaluate the effects of guar gum supplement for calcite precipitation enhanced by B. velezensis. The authors describe the idea for the experiment design in the introduction section. The concept for the selection of B. velezensis based on the positive traits of urease activity and calcium carbonate precipitation, which conducted good result on guar gum supplement is also briefly described in the introduction section. However, the idea are not stated clearly in the abstract.

2.Primers for 16S rRNA gene sequencing should have a reference cited. How about the similarity to the known Bacillus velezensis?

3.Figure titles are not necessary included in the results.

4.In the figure legends, bacterial strain should be indicated. Is the same for the experiments done? The consistence of materials used is very important.

Reviewer #2: This is a very interesting and relevant study for the MICP field. It is generally well-written and should be published after some minor changes.

Please clarify what “the green brick” refers to. Should it not be “a green brick”? Also, in the same sentence, it is likely the urea present in urine that is of interest and not the other nutrients. Please change and edit. Also, clarify what “it” refers to in the next sentence.

You mention “strains belonging to Bacillus…such as alkaline pH” but there is no reference for this. You therefore might want to add this reference: https://www.sciencedirect.com/science/article/abs/pii/S2213343718304251

Change “Since medium pH” to “Since the medium pH”

Screening for urea hydrolysis: it is not clear from this section how the concentration of urea and ammonium ions were measured once urea had hydrolysed. Please clarify.

“Further, qualitative analysis based on pH indicator (phenol red) for urea degradation” – was the urea degradation not quantified by measuring the concentration of ammonium ions, and if not, why not?

Fig S1, please confirm what the calcium concentration was and if this was the same for all bars shown in the graph.

“Aragonite was not found in precipitation in SMUC medium” – please mention why this might be the case.

Figure 6a: should it not be ammonium ions on the y-axis?

"...known to be the most stable form of calcium carbonate [39]" – it might be stable but it might also not be the strongest form. Please add some literature here about the strength of the different calcium carbonate types.

“guar gum as an economical substrate for bacterial cultivation” – mention is made of ‘economics’ but no economic value or cost of guar gum is given. Please add this or revise this sentence.

Reviewer #3: 1.Vaterite and calcite crystals are spherical and rhombohedral structure, respectively. Why are these structures not seen in the SEM micrographs? Sometimes, bacterium holes can be seen in the crystal surface, why not here?

2.¿Can you obtain percentages of vaterite and calcite using Rietveld refinement? It is important to know crystal proportion to evidence effect of guar gum in the process.

3.How was microbiological precipitation measured in the Figure S1?

4.Where is the microorganism shown to grow from 15 to 45 °C?

5.In the last paragraph of page 10, the conclusion “This may be the probable reason for higher ammonium ion concentration observed with medium supplemented with guar gum” should be revised. The higher ammonium ion concentration is due urea hydrolysis with or without guar gum, for that reason, the presence or formation of different detected metabolites should be explained better.

6.I recommended placing Figure 1 to supplementary material because it is not enough important to put in the principal page of the paper.

6. PLOS authors have the option to publish the peer review history of their article (what does this mean?). If published, this will include your full peer review and any attached files.

Reviewer #1: No

Reviewer #2: No

Reviewer #3: Yes: Sandra Patricia Chaparro Acuña

---

## [Author Response · Author response to Decision Letter 0]

25 Jun 2020

We have carefully examined these comments and introduced substantial changes and additions to the manuscript. The most significant revisions include the following: 

The abstract has been modified to enhance the clarity of the subject in manuscript.

Reference for the primers used in 16S rRNA gene sequencing for identification of isolated strain has been incorporated in the main manuscript.

As per the reviewer suggestion, figure 1 has been removed from the main manuscript and shifted in supplementary material and accordingly the numbering of the figures has been revised.

Previous Figure 3 has been split to figure 3 and figure 4. Relevant text has also been added in the result section under the subsection ‘Precipitation under flask condition and its microstructural analysis’ for better clarity on the Bacillus velezensis induced precipitation with and without guar gum supplementation.

Figure 5 has been modified and Table S3 is added to provide additional information obtained from XRD technique on polymorphic phases of calcium carbonate produced by experimented strain.

One figure S4 characterizing bacterial ability to grow under varied range of temperatures (15-45 °C) has been added in the supplementary material and relevant text is added in the manuscript in the result section under the subsection ‘Screening and identification of strains’

We believe, with these revisions the concerns of the reviewers have been adequately addressed.

Together with this letter we are submitting a point-by-point response to the reviewers’ comments.

---

## [Decision Letter · Decision Letter 1]

14 Jul 2020

Microbially induced calcite precipitation using Bacillus velezensis with guar gum

PONE-D-20-07309R1

Dear Dr. Kumar,

We’re pleased to inform you that your manuscript has been judged scientifically suitable for publication and will be formally accepted for publication once it meets all outstanding technical requirements.

Kind regards,

Arumugam Sundaramanickam, PhD

Academic Editor

PLOS ONE

Additional Editor Comments (optional):

Reviewers' comments:

Reviewer's Responses to Questions

**Comments to the Author**

1. If the authors have adequately addressed your comments raised in a previous round of review and you feel that this manuscript is now acceptable for publication, you may indicate that here to bypass the “Comments to the Author” section, enter your conflict of interest statement in the “Confidential to Editor” section, and submit your "Accept" recommendation.

Reviewer #2: All comments have been addressed

Reviewer #3: All comments have been addressed

2. Is the manuscript technically sound, and do the data support the conclusions?

Reviewer #2: Yes

Reviewer #3: Yes

3. Has the statistical analysis been performed appropriately and rigorously? 

Reviewer #2: Yes

Reviewer #3: Yes

4. Have the authors made all data underlying the findings in their manuscript fully available?

Reviewer #2: Yes

Reviewer #3: Yes

5. Is the manuscript presented in an intelligible fashion and written in standard English?

Reviewer #2: Yes

Reviewer #3: Yes

6. Review Comments to the Author

Reviewer #2: I am happy with the revised manuscript and comprehensive changes that have been made by the authors.

Reviewer #3: (No Response)

7. PLOS authors have the option to publish the peer review history of their article (what does this mean?). If published, this will include your full peer review and any attached files.

Reviewer #2: No

Reviewer #3: No

---

## [Editor Report · Acceptance letter]

17 Jul 2020

PONE-D-20-07309R1 

Microbially induced calcite precipitation using Bacillus velezensis with guar gum 

Dear Dr. Kumar:

I'm pleased to inform you that your manuscript has been deemed suitable for publication in PLOS ONE. Congratulations! Your manuscript is now with our production department. 

Kind regards, 

on behalf of

Professor Arumugam Sundaramanickam 

Academic Editor

PLOS ONE